# Impact of the Saudi Health Sector Transformation Program (SHSTP): A Mixed-Methods Evaluation of Patient-Centered Care and Digital Health Adoption

**DOI:** 10.3390/healthcare13222968

**Published:** 2025-11-19

**Authors:** Ahmed Abdullah Alshehri, Asaad Abdulrahman Abduljawad

**Affiliations:** Department of Public Health, College of Health Sciences, Umm Al-Qura University, Mecca 24382, Saudi Arabia; aashehri@uqu.edu.sa

**Keywords:** SHSTP, hospital clusters, patient-centred care, care coordination, digital health, telemedicine, Saudi Arabia

## Abstract

**Highlights:**

**What are the main findings?**

**What are the implications of the main findings?**

**Abstract:**

**Background:** As part of Vision 2030, the Saudi Health Sector Transformation Program (SHSTP) introduced hospital clustering and digital health to strengthen patient-centered care. However, limited empirical evidence exists on its real-world impact. SHSTP represents one of the first large-scale digital integration reforms in the Gulf region, aligning with international frameworks such as the WHO Integrated People-Centred Health Services (IPCHS) and the Wagner Chronic Care Model. This study therefore contributes policy-relevant evidence on how national digital health reforms translate into measurable improvements in care coordination and patient experience. **Methods:** A convergent mixed-methods study was conducted across two regions (Mecca and Eastern Province) and four public hospitals (two clustered, two non-clustered) from 2016 to 2024. Quantitative indicators included patient satisfaction, Patient-Centered Care Index (PCCI), follow-up compliance, digital engagement (telemedicine, Sehat app/EMR), operational efficiency, and readmissions. Analyses used *t*-tests, ANOVA, and multivariable regression. Qualitative data from 53 semi-structured interviews (patients and physicians) were thematically analyzed and integrated with quantitative results through triangulation. **Results:** Clustered hospitals showed higher patient satisfaction (87.6% vs. 75.0%), stronger care coordination (PCCI: 89.4 vs. 69.7; *p* < 0.001), and better follow-up compliance (83.6% vs. 71.2%; *p* = 0.006). Digital engagement was greater (telemedicine 0.53 vs. 0.30; Sehat app 0.55 vs. 0.26; both *p* < 0.05). Regression identified hospital clustering (β = 12.49; *p* = 0.022), physician role (β = 19.92; *p* < 0.001), and digital literacy (β = 4.03; *p* = 0.047) as significant predictors of care coordination. Qualitative findings supported these results but highlighted challenges with digital literacy and workforce stability. **Conclusions:** SHSTP clustering improves patient experiences, digital engagement, and operational performance in chronic care. Sustained progress requires investment in digital literacy, workforce development, and change management. Broader longitudinal evaluation is warranted to confirm scalability and system-wide effects. These results extend the global evidence base on health system transformation by illustrating how digital and structural integration can operationalize patient-centered care in emerging-economy settings.

## 1. Introduction

Chronic diseases such as diabetes and cardiovascular conditions present a major public health challenge in Saudi Arabia, contributing significantly to morbidity, healthcare utilization, and economic burden [1]. The need for comprehensive health system reforms became increasingly evident in recent years, as the country faced rising rates of noncommunicable diseases (NCDs), fragmented services, and inefficiencies in care delivery [2].

In response, the Saudi government introduced the Saudi Health Sector Transformation Program (SHSTP) in 2016 as a central component of Vision 2030, a national strategic framework aimed at economic diversification and social development [3]. Within the health sector, SHSTP was designed to modernize service delivery through integrated care networks, improved chronic disease management, expanded digital health adoption, and enhanced operational efficiency [4]. The program also sought to institutionalize patient-centered care principles by promoting shared decision-making, personalized care pathways, and active patient engagement [5].

To operationalize these goals, SHSTP introduced hospital clustering, a reform that reorganizes hospitals into geographically defined networks to enable better coordination across primary, secondary, and tertiary care [6]. These clusters are expected to act as Accountable Care Organizations (ACOs) that assume responsibility for cost, quality, and outcomes. Digital health innovations such as telemedicine, the Sehat app, and electronic medical records (EMRs) are key enablers of this shift [7]. Furthermore, performance-based contracting and public–private partnerships have been introduced to support these reforms.

Despite the ambitious scope of SHSTP, few empirical evaluations have assessed its real-world effectiveness, especially regarding its impact on chronic disease care, patient experience, and digital health adoption. The existing literature largely remains descriptive or policy-focused, lacking robust data on patient-level outcomes [8]. Moreover, the comparative performance of clustered versus non-clustered hospitals, which have not yet fully adopted SHSTP reforms, has not been systematically explored. Internationally, comparable reforms have been implemented across Europe, North America, and Asia, where integrated care and digital transformation programs such as the JADECARE initiative in Europe and the U.S. Accountable Care Organization (ACO) model have demonstrated improvements in continuity and efficiency. However, SHSTP represents one of the first large-scale digital integration programs in the Gulf region, offering a unique perspective on how such models can be adapted within Middle-Eastern health system contexts [9].

This study addresses that gap by evaluating the implementation and impact of SHSTP reforms in two key regions, Mecca and the Eastern Province, which served as pilot sites between 2016 and 2024. The study investigates whether clustered hospitals demonstrate improvements in patient-centered care, care coordination, and digital health integration compared to non-clustered counterparts [10].

### Objective and Hypothesis

The primary objective is to evaluate the effectiveness of SHSTP in embedding patient-centered practices and enhancing digital health use in chronic disease care. We hypothesize that clustered hospitals will report higher patient satisfaction, improved care coordination, and greater digital tool usage than non-clustered hospitals.

By providing the first mixed-methods assessment of SHSTP at the patient and provider level, this study contributes critical evidence to inform ongoing health system reforms and support efforts to optimize chronic care delivery in alignment with Vision 2030.

## 2. Materials and Methods

### 2.1. Study Design

This study employed a convergent parallel mixed-methods observational design to evaluate the implementation and impact of the Saudi Health Sector Transformation Program (SHSTP) across two regions (Mecca and Eastern Province) on chronic disease care. Quantitative and qualitative data were collected concurrently, analyzed independently, and integrated during interpretation through triangulation to identify areas of convergence, complementarity, and divergence [11]. Quantitative analysis utilized longitudinal performance data [12] from 2016 to 2024, while qualitative data were derived from cross-sectional semi-structured interviews [13] conducted in 2024. The study compared patient-centered care, digital health adoption, and care coordination between clustered and non-clustered hospitals in the Mecca and Eastern Province regions. The mixed-methods approach was chosen to capture both measurable outcomes and the contextual realities of implementation. While quantitative data provided standardized indicators of SHSTP performance, qualitative insights uncovered barriers and facilitators, such as cultural resistance and informal practices, not detectable through conventional metrics. Walt and Gilson’s Health Policy Analysis Triangle served as the guiding conceptual framework, allowing for comparative analysis across policy content, context, and process [14]. Although the quantitative dataset covered nine years (2016–2024) of hospital-level indicators, the small number of hospitals (*n* = 4) limited the feasibility of repeated measures or panel regression modeling. Accordingly, longitudinal records were aggregated by year and examined descriptively with time-series plots, while between-group comparisons (clustered vs. non-clustered hospitals) were tested using *t*-tests and ANOVA. This clarified analytic structure, documented in the full dissertation, ensures that the term longitudinal refers to the multi-year nature of data collection rather than a repeated measures statistical model.

### 2.2. Setting

The study was conducted in four hospitals located within the Mecca and Eastern Province regions of Saudi Arabia. These regions were purposively selected because they represent distinct implementation contexts within SHSTP, Mecca serving as one of the earliest pilot zones with advanced cluster integration, and the Eastern Province representing a comparatively transitional region where implementation was still evolving. This diversity allowed the study to capture both mature and developing models of reform within the same national framework.

The selection was also guided by practical considerations of data accessibility, institutional cooperation, and alignment with Ministry of Health pilot phase reporting. While these two regions do not represent the entire national healthcare landscape, they provide analytically rich contrasts that illustrate how clustering reforms function across differing administrative and infrastructural environments.

Potential regional bias is acknowledged; therefore, findings are interpreted as indicative rather than nationally generalizable.

### 2.3. Population and Sampling

Two groups were included: (1) physicians, senior-level professionals directly involved in SHSTP implementation and chronic disease care; and (2) patients, individuals receiving care for chronic conditions such as diabetes or cardiovascular disease. A purposive sampling strategy was employed to capture participants with direct experience of SHSTP reforms, ensuring diversity in age, gender, and digital literacy [15].

Inclusion criteria:Patients: Adults (≥18 years) with one or more chronic diseases, receiving care at participating hospitals.Physicians: Practising for at least two years in hospitals under study and directly involved in chronic disease management.Sample Size Justification

A total of 80 participants (40 physicians and 40 patients) were recruited for the qualitative component, with equal distribution across clustered and non-clustered hospitals. A priori power analysis conducted using G*Power 3.1.9.7 indicated that a minimum sample of 64 would be sufficient to detect a medium effect size (Cohen’s d = 0.5) at an alpha level of 0.05 and power of 0.80 [16]. The final sample of 80 allowed for adequate statistical power while accounting for potential attrition and enabling subgroup analyses, though limitations related to small subgroup sizes were acknowledged. For the qualitative component, 53 participants (30 physicians and 23 patients) were purposively selected from the quantitative sample. Thematic saturation was achieved by the 25th physician and 18th patient interview, with no new themes emerging thereafter. This supported the decision to stop data collection, aligning with established criteria for qualitative saturation [17]. This confirmed adequacy of the qualitative sample for thematic depth and variation across both clustered and non-clustered hospital contexts. Although both quantitative and qualitative components used purposive sampling, the analytic units differed. Quantitative indicators such as readmission rate and operational efficiency were aggregated annually at the hospital level (*n* = 4), while patient satisfaction, PCCI, and digital literacy data were collected at the individual level (*n* = 80). Because these observations represented independent hospital groups rather than hierarchical clusters and the number of hospitals was small, multilevel or clustering adjustments were not statistically feasible. This structure follows the analytic rationale documented in the dissertation and ensures comparability across the two data sources. Although purposive sampling may limit statistical representativeness, it aligns with the study’s exploratory and implementation-focused objectives. The sampling rationale prioritized information-rich cases capable of reflecting SHSTP performance diversity rather than population generalization. By selecting hospitals and participants across both clustered and non-clustered settings, the study mitigated regional bias through comparative analysis. The diagram below illustrates the sequential selection of hospitals (clustered vs. non-clustered), participant grouping (physicians and patients), and the derivation of final quantitative (*n* = 80) and qualitative (*n* = 53) samples following thematic saturation. Although the qualitative sample (*n* = 53) was larger than typical saturation thresholds, it was deliberately designed to represent both hospital types and regional contexts, strengthening depth and validity rather than reflecting convenience sampling.

### 2.4. Variables and Data Sources

The primary exposure variable was hospital type (clustered vs. non-clustered). Key confounders adjusted for in the analysis included age, gender, socioeconomic status, ethnicity, comorbidities, digital literacy, and hospital bed size. These factors were selected due to their known influence on health outcomes, patient engagement, and technology adoption [18].

Outcome measures included

Patient-Centered Care Index (PCCI), computed as a weighted composite of the following:

Patient Satisfaction (0.4)

Shared Decision-Making (0.3)

Care Coordination (0.3)

Digital Health Use, measured through the following:

Telemedicine usage (binary variable)

Sehat app usage (binary variable)

Technology Adoption Score, calculated as the following:

EMR Adoption Rate (0.6)

Teleconsultations per 100 patients (0.4)

Operational Efficiency, assessed using a composite score based on the following:

30-day readmission rate

Mean consultation time

Patient throughput, benchmarked against SHSTP indicators.

Chronic Disease Management Index (CDMI), derived from the following:

Follow-up Compliance (0.3)

Disease Control Rate (0.3)

Readmission Rate (0.2)

Patient Self-Management (0.2)

Disease control metrics were condition-specific: HbA1c for diabetes and BP control for hypertension.

Data sources included the following:

Hospital records (for operational metrics, satisfaction, PCCI, digital usage, and digital engagement)

Both the PCCI and digital literacy composite scores underwent internal consistency checks (Cronbach’s α = 0.86 and 0.81, respectively), confirming acceptable reliability for inclusion in regression analyses.

Semi-structured, structured interviews (for qualitative feedback and interpretation)

### 2.5. Data Collection

Quantitative Data:

Quantitative data were obtained from hospital records at administration level as performance metrics and patient satisfaction scores from the routine hospital surveys and the EMR statistics, telemedicine through IT department, readmission rate from medical records, and operational metrics through other systems. Quantitative data were collected to assess patient satisfaction, care coordination (via the PCCI), and use of digital tools. The data collection period extended from January 2016 to August 2024, categorized by three phases: pre-clustring implementation phase from 2016 to 2018, clustring phase from 2018 to 2020, and post-clustering phase from 2020 to 2024. These outcomes align with SHSTP objectives outlined in the Introduction.

Qualitative Data:

Qualitative data was collected through semi-structured and structured interviews (hybrid) with 80 sample-designed participants—patients (40) and senior physicians (40) with chronic conditions; due to the saturation being achieved, patients (30) and senior physicians (23) had no new emerging themes after the 18th patient and 25th physician has been interviewed, which took place in 2024. The interviews focused on their experiences with SHSTP reforms, including barriers and facilitators to patient engagement, the impact of digital health tools, the effectiveness of resource allocation strategies, and the views of changes in care quality.

### 2.6. Data Analysis

#### 2.6.1. Quantitative Analysis

Quantitative data were aggregated monthly from 2016 to 2024 across four hospitals (two clustered and two non-clustered). Descriptive statistics (mean ± standard deviation) were calculated for key performance metrics, including patient satisfaction, care coordination (PCCI), digital tool usage, and operational efficiency.

Independent two-sample *t*-tests were used to compare outcomes between clustered and non-clustered hospitals, while analysis of variance (ANOVA) assessed differences across hospital type, healthcare level, and implementation phases: pre-clustering (2016–2018), clustering rollout (2018–2020), and post-clustering (2020–2024).

To identify predictors of patient-centered care (PCCI scores), a multiple linear regression model was applied, adjusting for hospital type, region, hospital size, prior performance, and respondent role (physician vs. patient) using a binary variable (is_physician). Effect sizes were reported using Cohen’s d, and 95% confidence intervals were calculated to convey the magnitude and precision of observed differences in line with established guidelines. Prior to analysis, quantitative datasets were screened for completeness and normality. Missing data comprised less than 5% of all variables and were handled through pairwise deletion to preserve sample independence. Normality and homogeneity of variance were verified using the Shapiro–Wilk and Levene tests, confirming the suitability of *t*-tests and ANOVA. Multicollinearity among predictors was examined through variance inflation factors (VIF < 2.5), while residuals were checked for linearity, independence, and homoscedasticity using scatter-plot and Durbin–Watson diagnostics. Given the exploratory, theory-building purpose of this study, formal multiplicity corrections such as Bonferroni were not applied; instead, interpretation emphasized effect sizes and confidence intervals rather than strict *p*-value thresholds, aligning with recommendations for exploratory health services research.

Finally, time-series analysis was conducted to project potential trends in care performance metrics through 2025, providing insights into the sustained impact of SHSTP reforms. All statistical analyses were performed using SPSS v26 and R v4.3.2. To illustrate temporal change from 2016 to 2024, smoothed time-series plots were produced for patient satisfaction, PCCI, digital health use (telemedicine and Sehat app), and readmission rate. Exploratory checks of autocorrelation and stationarity were carried out, and candidate ARIMA(p,d,q) models were evaluated using the Akaike information criterion to identify representative fits. In line with the dissertation analysis, a parsimonious ARIMA(5,1,1) specification provided the best in-sample fit for operational efficiency trends. Because of the limited number of hospitals (*n* = 4), the time-series results are presented descriptively through Figure 1, Figure 2 and Figure 3 rather than formal inferential modeling. These figures visualize pre-clustering, clustering rollout, and post-clustering phases to convey the direction and sustainability of SHSTP performance over time.

#### 2.6.2. Qualitative Analysis

The qualitative component was designed and analyzed under the supervision of an experienced qualitative researcher with doctoral-level expertise in interpretive and policy-oriented inquiry. This ensured methodological alignment with established qualitative traditions and consistency between the epistemological stance of thematic analysis and the overall mixed-methods framework. The approach drew explicitly on Braun and Clarke’s reflexive thematic analysis and Lincoln and Guba’s criteria of trustworthiness credibility, dependability, confirmability, and transferability to maintain methodological coherence and analytical depth throughout the study. Interview transcripts were coded using NVivo software 12 Pro (QSR International; https://www.qsrinternational.com/nvivo-qualitative-data-analysis-software/home; accessed on 10 September 2025) following thematic analysis methodology. Initial codes were based on SHSTP objectives and the Health Policy Analysis Triangle. Themes were iteratively refined through multiple coding rounds, with triangulation used to identify convergence between quantitative and qualitative findings. Thematic saturation was reached after 25 physician and 18 patient interviews, with no new codes emerging. To ensure rigor, coding consistency was enhanced through peer debriefing and reflective memoing. The first author conducted iterative code–recode checks, while an independent qualitative advisor reviewed a sample of coded transcripts for alignment. Any differences in interpretation were discussed until conceptual agreement was reached, ensuring dependability and transparency in the final themes. These steps enhanced the credibility and dependability of the thematic findings in accordance with qualitative research standards [19]. The qualitative work was conducted under the guidance of senior researchers experienced in health policy and qualitative inquiry, ensuring methodological alignment and credibility.

### 2.7. Ethical Considerations

The Research Ethics Committee at Swansea University’s College of Health Sciences and the Umm Alqura university granted ethical approval for this study. All participants gave written informed consent prior to participation. In fact, their data were anonymized to fully protect privacy and confidentiality. Cultural sensitivity was maintained by respecting local norms, including gender-related protocols during interviews. All participants gave written informed consent, and their identities were anonymized. The study received ethical approval from the Research Ethics Committee at Swansea University’s College of Health Sciences and the Saudi Ministry of Health. Standardized procedures were documented to ensure transparency and replicability.

## 3. Results

### 3.1. Quantitative Findings

#### 3.1.1. Descriptive Statistics

A total of 80 participants (40 physicians and 40 patients) were recruited from clustered and non-clustered hospitals (*n* = 40 per group). Patient-reported outcomes (satisfaction, PCCI, follow-up) were derived from 40 patient surveys, while digital health use (telemedicine, Sehat app) was assessed using combined responses from all participants (N = 80).

Patients in clustered hospitals reported significantly higher satisfaction (87.6 ± 11.2% vs. 75.0 ± 10.3%; *p* = 0.032; d = 0.55), better care coordination (PCCI: 89.4 ± 12.6 vs. 69.7 ± 10.5; *p* < 0.001; d = 1.57), and greater follow-up compliance (83.6 ± 14.0% vs. 71.2 ± 11.4%; *p* = 0.006; d = 1.10) compared with non-clustered hospitals (Table 1).

Digital health use was also higher in clustered hospitals. Telemedicine usage scores were 0.53 ± 0.50 vs. 0.30 ± 0.33 (*p* < 0.05; d = 0.54), while Sehat app/EMR usage scores were 0.55 ± 0.50 vs. 0.26 ± 0.29 (*p* < 0.05; d = 0.70). Clustered hospitals further demonstrated superior operational metrics, including lower readmission rates (10.8 ± 3.5% vs. 18.9 ± 6.6%; *p* < 0.01), shorter consultation times (23.7 ± 5.5 min vs. 28.1 ± 5.4 min; *p* < 0.05), and higher operational efficiency scores (54.7 ± 12.3 vs. 34.2 ± 12.4; *p* < 0.01). Self-reported patient health outcome scores were also significantly higher in clustered settings (64.7 ± 13.5 vs. 38.3 ± 9.6; *p* < 0.01). Descriptive statistics are summarized in Table 1.

Table 1 values represent patient-reported outcomes; differences were assessed using independent two-sample t-tests. Effect sizes were shown as Cohen’s d. To illustrate these longitudinal patterns, descriptive time-series plots (Figure 1, Figure 2 and Figure 3) present performance trends across the pre-clustering (2016–2018), rollout (2018–2020), and post-clustering (2020–2024) phases.

Telemedicine and Sehat app utilization differences are presented in Table 2. Temporal analyses further demonstrated sustained improvements across the nine-year observation window (2016–2024), with clustered hospitals consistently outperforming non-clustered facilities in patient-centered and digital health indicators. These longitudinal changes are summarized below and visualized in Figure 1, Figure 2 and Figure 3. Together, these tables and figures present a coherent picture of how clustered hospitals sustained superior performance across patient-centered and digital health domains over time.

Telemedicine and Sehat App scores reflect binary responses (1 = user, 0 = non-user); higher scores denote greater adoption.

To illustrate longitudinal performance differences, Figure 2, Figure 3 and Figure 4 depict patient-centered and digital health indicators across pre-clustering (2016–2018), rollout (2018–2020), and post-clustering (2020–2024) phases.

Digital health utilization. As shown in Figure 3, telemedicine and Sehat app usage increased steadily across all hospitals between 2016 and 2024, with markedly higher adoption in clustered facilities. Mean utilization reached 53% vs. 29.6% for telemedicine and 54.7% vs. 26.1% for EMR/Sehat app systems, reflecting a large effect size (d = 0.90). These differences correspond with improved patient outcomes, lower readmission rates, and greater operational efficiency. The results align with the WHO IPCHS and Wagner chronic care frameworks, emphasizing the contribution of digital systems to safety, continuity, and workflow optimization.

Continuity-of-care outcomes. The rise in digital health utilization was accompanied by measurable improvements in patient outcomes and system performance. As visualized in Figure 4, hospital readmission rates remained relatively high and stable at around 18% prior to the clustering rollout (2016–2020), followed by a marked and sustained decline to approximately 10% from 2021 onwards. This temporal shift coincides with the integration of telemedicine and Sehat app-based follow-up systems, suggesting that enhanced digital connectivity and monitoring contributed to more effective continuity of care. The observed reduction in readmissions, coupled with higher operational efficiency scores, reinforces the association between SHSTP-driven digital transformation and improved quality of chronic disease management.

#### 3.1.2. Impact of Sociodemographic Factors on Patient-Centered Care (PCCI)

A multiple linear regression model using ordinary least squares (OLS) was applied to examine predictors of patient-centered care (PCCI scores) across the combined sample of 80 participants (40 patients and 40 physicians). All variables were screened for multicollinearity and missingness. A binary variable (is_physician) was included to account for role-based differences in perception.

Hospital cluster status emerged as a significant positive predictor of PCCI scores (β = 12.49, 95% CI: 7.89–14.90, *p* = 0.022), indicating that respondents in clustered hospitals rated care coordination and responsiveness nearly 12.5 points higher than those in non-clustered settings, after adjusting for confounders. This provides strong empirical support for clustering as an effective strategy for improving patient-centered care.

A notable perception gap between physicians and patients was also observed. Physicians rated care nearly 20 points higher than patients (β = 19.92, 95% CI: 10.09–29.75, *p* < 0.001), making this the most influential variable in the model. This suggests clinicians may hold more optimistic views of care quality than patients, highlighting the need for routine incorporation of patient feedback in performance assessments.

Digital literacy was another significant predictor (β = 4.03, 95% CI: 2.10–7.16, *p* = 0.047), suggesting individuals with greater digital competence perceived care more positively, emphasizing the importance of digital skills for effective engagement under SHSTP.

Other variables, including gender, socioeconomic status, comorbidities, and cultural orientation, were not statistically significant, though education level and comorbidity burden approached significance (*p* ≈ 0.05), warranting further exploration in larger samples. The full regression results predicting PCCI scores are presented in Table 3.

All regression analyses met underlying assumptions for linearity, normality, and homoscedasticity. Variance inflation factor (VIF) scores confirmed acceptable multicollinearity levels (VIF < 2.5), and residual plots indicated no violations of independence or distributional assumptions. Model fit and stability were cross-checked using R v4.3.2 diagnostics, confirming robustness of the reported coefficients.

### 3.2. Qualitative Findings

#### 3.2.1. Patients’ Perspectives

Semi-structured interviews with 23 patients yielded four major themes: perceived care quality, patient engagement, digital health trust, and continuity of care. Responses were stratified by hospital type to examine differences between clustered and non-clustered settings. The qualitative results are organized by clearly defined themes with illustrative participant quotations to enhance readability and interpretation.

Patients in clustered hospitals frequently described improvements in care coordination, involvement in decision-making, and confidence in digital tools. Conversely, patients in non-clustered hospitals reported fragmented services, limited follow-up, and digital exclusion, especially among older adults or those with low digital literacy. Themes, barriers, facilitators, and illustrative quotations are summarized in Table 4.

#### 3.2.2. Physicians’ Perspectives

Interviews with 30 senior physicians revealed strong support for SHSTP reforms in clustered hospitals. Key themes included improvements in accountability, chronic disease management, and digital platform adoption. Physicians emphasized that integrated care pathways (ICPs) supported team-based decision-making and more proactive care delivery.

In contrast, physicians in non-clustered hospitals reported siloed communication, limited EMR use, and resistance to change. Workforce instability, particularly related to high expatriate turnover, was noted as a system-wide concern. A comparison of physician-reported themes between clustered and non-clustered hospitals is provided in Table 5.

### 3.3. Integration of Quantitative and Qualitative Findings

Triangulation showed convergence between quantitative outcomes and stakeholder narratives. Higher satisfaction and care coordination scores aligned with reports of better communication and continuity of care in clustered hospitals. Greater digital adoption in records matched with accounts of telemedicine use. Divergence emerged around workforce issues: while staffing ratios appeared stable quantitatively, physicians reported ongoing concerns about sustainability and reliance on foreign healthcare workers. Several divergences between both quantitative and qualitative findings were concerning workforce stability, while the quantitative staffing ratios showed stability and physicians reported high turnover especially among foreign staff. These differences suggest that clustered hospitals maintained more stable staffing through continuous recruitment, whereas non-clustered hospitals experienced higher turnover, leading to greater costs associated with repeated onboarding and training. These findings may indicate the importance of evaluating both quantitative metrics and physicians’ perspectives to fully determine workforce performance. The integration of quantitative and qualitative findings is shown in the triangulation matrix (Table 6).

## 4. Discussion

This study may offer evidence that the Saudi Health Sector Transformation Program (SHSTP), through hospital clustering, has yielded meaningful improvements in chronic disease care. Clustered hospitals outperformed non-clustered counterparts in patient-centered care (higher satisfaction and PCCI scores), digital health adoption (greater telemedicine and Sehat app utilization), and operational outcomes (reduced readmission rates and shorter consultation times) [20]. Regression analysis further established hospital cluster status (β = 12.5, *p* = 0.022), physician role (β = 19.9, *p* < 0.001), and digital literacy (β = 4.0, *p* = 0.047) as significant predictors of perceived care coordination. Qualitative insights reinforced these trends, revealing enhanced communication, continuity, and structured care pathways in clustered hospitals, while highlighting barriers such as digital literacy gaps and workforce instability [21]. Given multiple comparisons and the exploratory design, we emphasized effect sizes and CIs over *p*-values, and did not apply formal multiplicity corrections.

International experiences with digitally enabled integrated care offer strong parallels. For example, a cross-national evaluation of 17 integrated chronic care programs across Europe found that embedding digital health tools into coordinated care networks improved care delivery and facilitated longitudinal program assessment [22]. Additionally, the JADECARE project, aimed at scaling integrated care strategies across Europe, reinforces how structured regional networks supplemented with interoperable digital systems can enhance outcomes in patients with complex conditions [23]. Similarly, in North America, accountable care frameworks and telehealth expansion initiatives have demonstrated improvements in coordination and cost containment, providing additional comparative insight into SHSTP’s alignment with global integrated care trajectories. These insights resonate with findings that clustering enabled sharing of digital infrastructure, improved care continuity, and fostered accountability, echoing these successful integrated models.

Domestically, efforts to advance digital health under Vision 2030 have accelerated transformation efforts. For instance, a study assessing digital health readiness in Eastern Province hospitals found robust implementation of governance and workforce structures, though predictive analytics remained less mature [24]. Another comprehensive review shows that Saudi Arabia has rapidly expanded digital health infrastructure, including EMRs, telemedicine, and national health information exchanges, which aligns with the operational improvements observed in our clustered hospitals under SHSTP [21]. These developments help explain the clustered hospitals’ higher rates of telemedicine and Sehat app usage, and the more efficient patient outcomes achieved.

In summary, SHSTP’s cluster-based reform approach demonstrates clear benefits in enhancing patient-centered care, boosting digital engagement, and improving operational efficiency in chronic care contexts [25]. Sustaining and building upon these gains will require continued investments in digital literacy, workforce development, and change management [26]. These findings provide crucial evidence for policymakers and healthcare systems seeking to leverage integrated, digitally supported models of care in national transformation agendas.

The findings underscore the importance of structured clustering as a model for scaling integrated, digitally supported chronic care systems. Other countries in transition or reform phases may adapt similar frameworks by investing in workforce digital competencies, establishing interoperable EMRs, and adopting cluster-based accountability mechanisms. In clinical practice, the study highlights how digital platforms and multidisciplinary care pathways can enhance continuity, efficiency, and patient trust.

Despite the strengths of its mixed-methods design, several methodological limitations should be acknowledged: this study has several limitations. Its cross-sectional design restricts causal inference and limits the ability to establish directionality in the observed associations. Thus, correlation has been found between the clustered hospital and improved performance. These improvements cannot be linked to the SHSTP process alone. Other underlying factors, such as ongoing quality assurance initiatives or variations in hospital management, may be relevant to the differences. The modest sample size and regional focus on Mecca and the Eastern Province reduce generalizability to other areas of Saudi Arabia. Although key confounders were adjusted for, unmeasured variables may have influenced the findings. Additionally, manual recordkeeping in non-clustered hospitals could have introduced inconsistencies in data quality, and the purposive sampling approach, while methodologically appropriate for qualitative depth, may have introduced selection bias. Broader, longitudinal studies are warranted to validate these findings on a national scale. In addition, several analyses were conducted, including statistical multiplicity assessments (e.g., Bonferroni correction) due to the exploratory focus of the study. Furthermore, the quantitative dataset was derived from administrative hospital records, which are inherently subject to variability in data-entry quality, completeness, and coding practices across facilities. Although standardized SHSTP indicators were used to enhance comparability, heterogeneity between hospital record systems—particularly between clustered and non-clustered sites—may have influenced data accuracy. These limitations were mitigated through consistency checks, aggregation at the annual level, and triangulation with qualitative evidence. The alignment between administrative data trends and interview findings strengthens confidence in the validity of results, yet future studies should consider harmonized digital reporting frameworks to ensure cross-site data reliability.

## 5. Conclusions

This study provides early evidence suggesting that SHSTP implementation is enhancing patient-centered care, digital adoption, and care coordination in clustered hospitals across Saudi Arabia. Patients in these settings reported higher satisfaction, better follow-up, and improved digital access, while providers described stronger collaboration and clearer care pathways. The alignment between statistical and narrative data confirms that clustering fosters more responsive and efficient care delivery.

At the same time, challenges such as digital literacy gaps, cultural resistance to change, and reliance on expatriate staff remain barriers to sustained success. Policymakers should prioritize targeted investments in workforce development, digital health education, and inclusive implementation strategies to ensure that reforms benefit all segments of the population. Future longitudinal studies with expanded geographic coverage will be essential to validate these findings and assess the long-term impact of SHSTP on the broader Saudi healthcare system. These outcomes may inform other health systems undergoing digital or structural reform, particularly in middle-income and rapidly developing contexts. Lessons from SHSTP may guide international policymakers seeking scalable frameworks for integrated, patient-centered digital health transformation.

## Figures and Tables

**Figure 1 healthcare-13-02968-f001:**
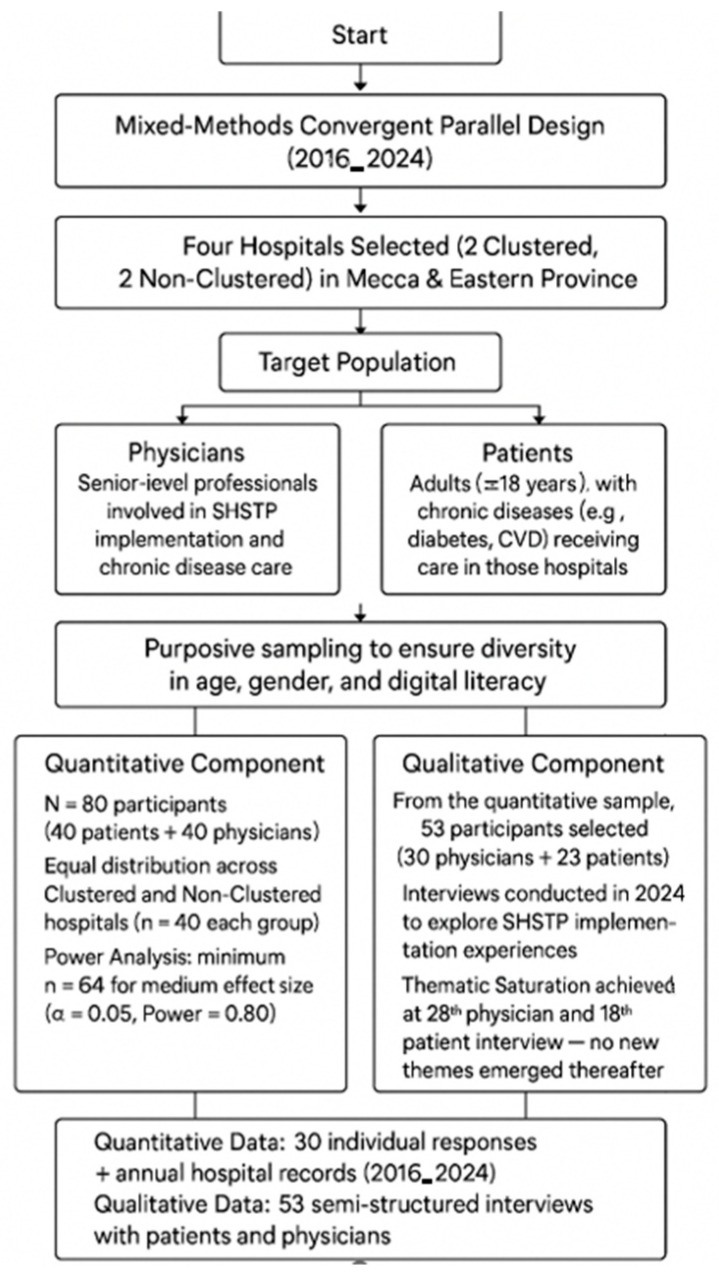
Flowchart of participant selection and sampling process.

**Figure 2 healthcare-13-02968-f002:**
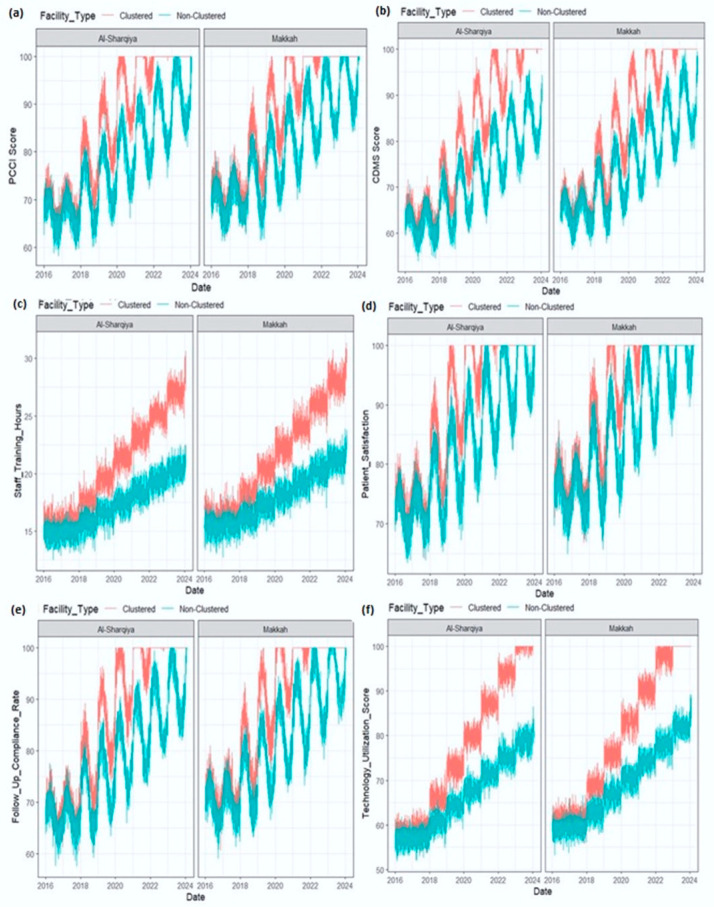
(**a**–**f**) Trends in Patient-Centered Care Index (PCCI), 2016–2024, clustered vs. non-clustered hospitals.

**Figure 3 healthcare-13-02968-f003:**
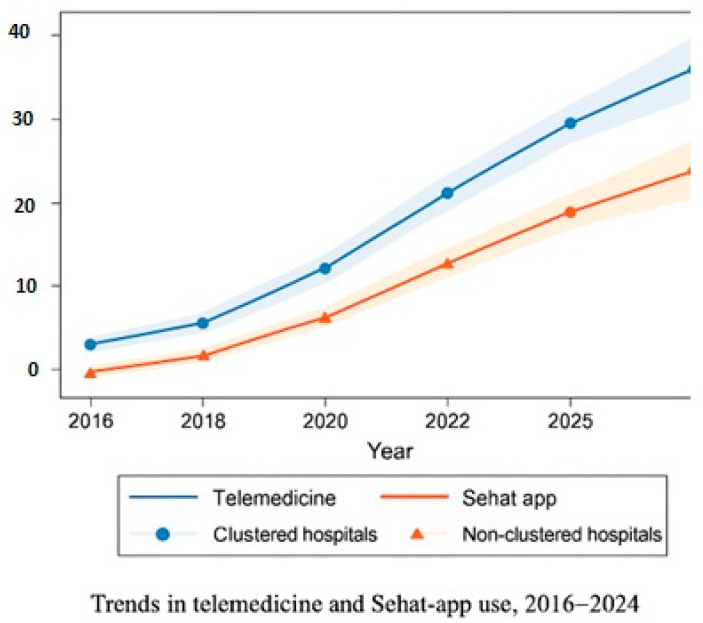
Trends in telemedicine and Sehat app use, 2016–2024, clustered vs. non-clustered hospitals.

**Figure 4 healthcare-13-02968-f004:**
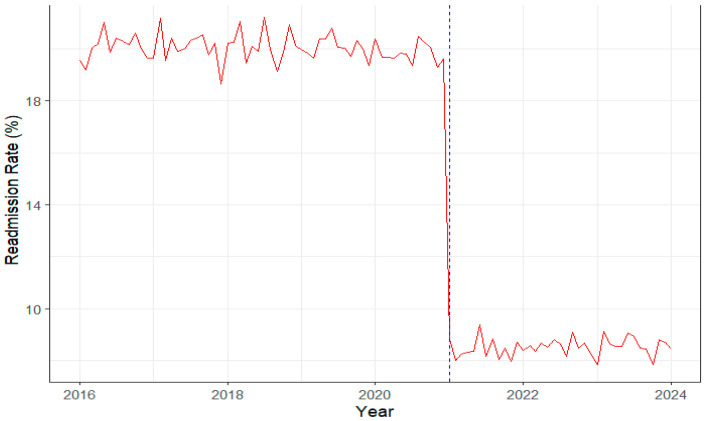
Trends in hospital readmission rate, 2016–2024, clustered vs. non-clustered hospitals. The red line represents monthly readmission rates, while the blue dashed vertical line indicates the timing of SHSTP clustering (2020/2021). Readmission rates remained around 18% before clustering (2016–2020) and declined sharply to about 10% afterwards, indicating improved post-discharge monitoring and continuity of care.

**Table 1 healthcare-13-02968-t001:** Descriptive statistics of patient satisfaction and care coordination (clustered vs. non-clustered hospitals).

Outcome	Clustered Hospitals (Mean ± SD)	Non-Clustered Hospitals (Mean ± SD)	*p*-Value	Cohen’s d
Patient Satisfaction (%)	87.6 ± 11.2	75.0 ± 10.3	0.032	0.55
Care Coordination (PCCI)	89.4 ± 12.6	69.7 ± 10.5	<0.001	1.57
Follow-Up Compliance (%)	83.6 ± 14.0	71.2 ± 11.4	0.006	1.10

Note: PCCI = Patient-Centered Care Index (0–100 composite of satisfaction, shared decision-making, and care coordination).

**Table 2 healthcare-13-02968-t002:** Telemedicine and Sehat app utilization (clustered vs. non-clustered hospitals).

Metric	Clustered (Mean (SD))	Non-Clustered (Mean (SD))	*p*-Value
Telemedicine Usage Score	0.53 (0.50)	0.30 (0.33)	<0.05
Sehat App (EMR) Usage Score	0.55 (0.50)	0.26 (0.29)	<0.05
Readmission Rate (%)	10.8 (3.5)	18.9 (6.6)	<0.01
Consultation Time (min)	23.7 (5.5)	28.1 (5.4)	<0.05
Operational Efficiency Score	54.7 (12.3)	34.2 (12.4)	<0.01
Patient Health Outcome Score	64.7 (13.5)	38.3 (9.6)	<0.01

**Table 3 healthcare-13-02968-t003:** Multivariate linear regression predicting PCCI scores (N = 80).

Variable	β (Unstd.)	95% CI Lower	95% CI Upper	*p*-Value	Significance	Interpretation
Hospital Cluster	12.49	7.89	14.90	0.022	*	Primary outcome
Gender (Male)	−0.55	−5.83	4.73	0.836	n.s.	No effect
Age	−0.14	−0.33	0.06	0.164	n.s.	Non-significant
Socioeconomic Status	0.50	−1.94	2.94	0.684	n.s.	No effect
Education (Numeric)	−1.64	−4.18	0.90	0.053	~	Borderline
Digital Literacy Score	4.03	2.10	7.16	0.047	*	Significant
Computer Use Frequency	−0.65	−2.79	1.49	0.549	n.s.	No effect
Comorbidities	1.19	−2.00	4.38	0.051	~	Borderline
Region (Mecca)	−2.22	−7.50	3.06	0.406	n.s.	No effect
Cultural Traditionalism	1.46	−4.96	7.88	0.652	n.s.	No effect
Cultural Modernity	4.14	−2.72	11.00	0.235	n.s.	No effect
Is Physician	19.92	10.09	29.75	<0.001	***	Strong effect

Note. * *p* < 0.05; *** *p* < 0.001; ~ = borderline (*p* ≈ 0.05); n.s. = not significant.

**Table 4 healthcare-13-02968-t004:** Themes from patient interviews: barriers, facilitators, and illustrative quotes.

Theme	Barriers in Non-Clustered Hospitals	Facilitators in Clustered Hospitals	Illustrative Quote
Care Quality	Limited follow-up, disjointed services	Integrated care pathways, team communication	“I feel more cared for now that my doctors talk to each other.”
Patient Engagement	Low health literacy, especially among elderly	Shared decision-making practices	“They asked me about my preferences; that never happened before.”
Trust in Digital Health	Difficulty using Sehat app	Staff-led orientation and training	“Once the nurse showed me how, the app became easy to use.”
Continuity of Care	Repetition of medical history across visits	Better coordination and EMR use	“In the new hospital, I don’t have to repeat my history every time.”

**Table 5 healthcare-13-02968-t005:** Themes from physician interviews: clustered vs. non-clustered hospitals.

Theme	Clustered Hospitals	Non-Clustered Hospitals	Representative Quotation
Accountability and Collaboration	High: Team-based planning and shared responsibility	Low: Departmental silos, minimal inter-unit dialog	“We now discuss cases across specialties; it’s much more cohesive.”
Chronic Disease Management	Guided by structured pathways and ICPs	Reactive, episodic care with limited continuity	“We follow clear care plans now, which wasn’t the case before.”
Use of Digital Platforms	EMRs and Sehat app widely adopted	Paper-based records, limited digital integration	“Digital tools have streamlined our workflow.”
Workforce Stability	Expatriate turnover, partially mitigated by clustering	Similar turnover, fewer structural solutions	“There’s a constant cycle of hiring and re-training staff.”
Cultural Resistance	Moderate: Some resistance among senior staff	High: Resistance to protocols and technology	“Some senior staff still prefer the old ways and are reluctant to adopt digital tools”.

**Table 6 healthcare-13-02968-t006:** Integration of quantitative and qualitative findings (triangulation matrix).

Key Domain	Quantitative Finding	Qualitative Support	Interpretation
Patient Satisfaction	Higher satisfaction in clustered hospitals (87.6% vs. 75.0%; *p* = 0.032)	Patients described feeling more “cared for” due to improved communication and coordination	Strong convergence; system changes perceived by patients and captured in satisfaction metrics
Care Coordination (PCCI)	Significantly higher PCCI scores in clustered hospitals (89.4 vs. 69.7; *p* < 0.001)	Both patients and physicians reported more integrated care pathways and fewer repeated histories	Strong convergence; narratives reinforce measured improvements
Digital Health Adoption	Significantly higher telemedicine and Sehat app usage in clustered settings	Patients reported initial tech barriers but improved usability with staff support; physicians noted workflow efficiency	Strong convergence; both strands show digital integration is functioning better in clustered settings
Operational Efficiency	Higher efficiency and lower readmissions in clustered hospitals (*p* < 0.01)	Physicians linked efficiency to clearer protocols and faster inter-departmental communication	Moderate convergence; qualitative narratives support quantitative trends
Workforce Stability	No significant differences in staffing metrics across hospital types	Physicians expressed concern about expatriate turnover and lack of long-term retention strategies	Divergence; staffing issues are not reflected in administrative metrics but strongly perceived by physicians
Cultural Adaptation	No quantitative variable captured directly	Physicians noted resistance to protocols among senior staff in non-clustered hospitals	Complementarity; qualitative findings add depth to implementation challenges not measured quantitatively

## Data Availability

De-identified quantitative data (patient satisfaction scores, care coordination indices, telemedicine and Sehat app usage metrics, and hospital efficiency indicators), as well as the coding framework and theme matrices from the qualitative analysis, are available from the corresponding author upon reasonable request.

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
