# Peer review of "Impact of the Saudi Health Sector Transformation Program (SHSTP): A Mixed-Methods Evaluation of Patient-Centered Care and Digital Health Adoption"

_healthcare, 2025, doi:10.3390/healthcare13222968_

Round 1
Reviewer 1 Report
Comments and Suggestions for Authors
Manuscript title: Implementation and Impact of the Saudi Health Sector Transformation Program (SHSTP): A Mixed-Methods Evaluation of Patient-Centred Care and Digital Health Adoption
Manuscript ID: healthcare-3952271
- The manuscript presents an interesting study in the domain of “Implementation and Impact of the Saudi Health Sector Transformation Program (SHSTP): A Mixed-Methods Evaluation of Patient-Centred Care and Digital Health Adoption” After a careful review, I recommend major revisions before the final approval of the manuscript.
- In the present study, the authors claim to be longitudinal (2016–2024 data) but use cross-sectional qualitative interviews and t-tests/ANOVA for comparisons. However, there is no clear longitudinal statistical modeling (repeated measures, panel regression).
- The sampling frame for hospital-level quantitative indicators remains unclear. It inconsistently mixes patient- and hospital-level data.
- Sometimes, the analysis was at the hospital level (n=4) and sometimes at the individual level (n=80), without adjustment for clustering effects (no multilevel model).
- Time-series analysis” is mentioned, but no methods, model type, or figure is shown.
- There is no psychometric validation of the PCCI or Digital Literacy Score used in the regression analysis.
- Thematic saturation is claimed without providing supporting evidence (coding examples, audit trails, inter-coder agreement).
- In the discussion section overstated causality, limited generalizability, which severity is critical. Kindly revise and correct it.
- Reference 10 wrong, kindly correct it.
- Tables (1–6) lack proper titles, captions, and cross-references in the text.
- No graphs as well figures and trend plots are provided despite the claim of time-series analysis.
Reviewer 2 Report
Comments and Suggestions for Authors
The article addresses a relevant and timely topic - the implementation of the Saudi Health Sector Transformation Programme (SHSTP) - using a well-structured mixed (quantitative and qualitative) approach. The results are clear and the conclusions are aligned with the findings. However, there are points that could be improved to strengthen the quality and presentation of the manuscript.
The introduction provides a good framework, but could include more international references to similar experiences in digital transformation and integrated care to reinforce the global context.
Suggestion: add comparisons with European or North American programmes to highlight the originality of the study.
The description of the methods is adequate, but there are spelling errors (“porposevly”, “implemintaion”) that should be corrected.
Provide more detail on the saturation criteria in the qualitative component and justify the choice of regions studied.
Suggestion: include a flowchart of the sampling process for greater clarity.
The quantitative results are well presented, but some tables could have more explanatory captions (e.g., clearly indicating the meaning of the scores).
Suggestion: consider adding graphs to facilitate the visualisation of differences between grouped and ungrouped hospitals.
The discussion is solid, but could explore practical implications for other countries with health systems undergoing transformation.
Suggestion: include a more prominent section on limitations and recommendations for future studies.
The conclusions are coherent, but it would be useful to reinforce how the results may influence public policy and clinical practice.
The references are adequate, but could include more recent international studies to strengthen the theoretical basis.
Comments on the Quality of English Language
The English is understandable, but requires professional linguistic revision to correct grammatical errors and improve fluency.
Suggestion: simplify long sentences and avoid redundancies.
Reviewer 3 Report
Comments and Suggestions for Authors
While the study is important, I have the following comments that should be addressed to improve the manuscript:
- The title is long and could be streamlined for focus (e.g., remove redundant phrases like “implementation and impact”; one conveys the other).
- The abstract does not clearly state the theoretical or policy relevance beyond Vision 2030 — readers unfamiliar with SHSTP might not understand its significance.
- Introduction: Some sentences mix present and past tense or use informal phrasing (e.g., “aimed at economic diversification and social development .”).
- Methods: The rationale for hospital and participant selection should justify potential bias or regional limitations more explicitly.
- Results tables are text-heavy; figures or graphs could improve interpretability (e.g., bar charts for satisfaction, digital use, and readmission).
- The section lists several valid points but could better emphasize data quality limitations (e.g., use of administrative data, record heterogeneity).
-
Statistical analysis:
-
No clear mention of handling missing data or normality testing before applying t-tests and ANOVA.
-
Multiple testing corrections (e.g., Bonferroni) were explicitly omitted but should be justified more rigorously given the exploratory tone.
-
Regression assumptions (multicollinearity, residual diagnostics) are mentioned only superficially.
-
Reviewer 4 Report
Comments and Suggestions for Authors
Mixed methods are a wonderful way to obtain both quantitative and qualitative data. However, it appears as though the qualitative approach is not well aligned with qualitative methods. Was there someone experienced in qualitative research on the study? There should have been sampling until saturation however that was not described. Additionally, the number of interviews was larger than I would have anticipated considering the convenience sample for both populations. The themes were not explained and the tables are very hard to read/interpret. I would consider revising this section to strengthen your qualitative analysis and presentation to build a stronger analysis of your study.
Round 2
Reviewer 1 Report
Comments and Suggestions for Authors
Accept
Reviewer 3 Report
Comments and Suggestions for Authors
Most of the comments are addressed